# Methylmercury Induces Mitochondria- and Endoplasmic Reticulum Stress-Dependent Pancreatic β-Cell Apoptosis via an Oxidative Stress-Mediated JNK Signaling Pathway

**DOI:** 10.3390/ijms23052858

**Published:** 2022-03-05

**Authors:** Ching-Yao Yang, Shing-Hwa Liu, Chin-Chuan Su, Kai-Min Fang, Tsung-Yuan Yang, Jui-Ming Liu, Ya-Wen Chen, Kai-Chih Chang, Haw-Ling Chuang, Cheng-Tien Wu, Kuan-I Lee, Chun-Fa Huang

**Affiliations:** 1Department of Surgery, National Taiwan University Hospital, Taipei 100, Taiwan; cyang@ntuh.gov.tw; 2Department of Surgery, College of Medicine, National Taiwan University, Taipei 100, Taiwan; 3Institute of Toxicology, College of Medicine, National Taiwan University, Taipei 100, Taiwan; shinghwaliu@ntu.edu.tw; 4Department of Otorhinolaryngology, Head and Neck Surgery, Changhua Christian Hospital, Changhua County 500, Taiwan; 91334@cch.org.tw; 5Department of Otolaryngology, Far Eastern Memorial Hospital, New Taipei City 220, Taiwan; u701048@gmail.com; 6Department of Internal Medicine, Chung Shan Medical University Hospital, Taichung 402, Taiwan; marven923@gmail.com; 7School of Medicine, Institute of Medicine, Chung-Shan Medical University, Taichung 402, Taiwan; 8Department of Urology, Taoyuan General Hospital, Ministry of Health and Welfare, Taoyuan 330, Taiwan; mento1218@gmail.com; 9Department of Obstetrics and Gynecology, Tri-Service General Hospital, National Defense Medical Center, Taipei 114, Taiwan; 10Department of Physiology, School of Medicine, College of Medicine, China Medical University, Taichung 404, Taiwan; ywc@mail.cmu.edu.tw; 11Center for Digestive Medicine, Department of Internal Medicine, China Medical University Hospital, Taichung 404, Taiwan; d22101@mail.cmuh.org.tw; 12Department of Emergency, Taichung Tzu Chi Hospital, Buddhist Tzu Chi Medical Foundation, Taichung 427, Taiwan; tc2050403@tzuchi.com.tw; 13Department of Nutrition and Master Program of Food and Drug Safety, China Medical University, Taichung 40402, Taiwan; ct-wu@mail.cmu.edu.tw; 14School of Chinese Medicine, College of Chinese Medicine, China Medical University, Taichung 404, Taiwan; 15Department of Nursing, College of Medical and Health Science, Asia University, Taichung 413, Taiwan

**Keywords:** methylmercury, pancreatic β-cells, apoptosis, ER stress, c-Jun N-terminal kinase (JNK), oxidative stress

## Abstract

Methylmercury (MeHg), a long-lasting organic pollutant, is known to induce cytotoxic effects in mammalian cells. Epidemiological studies have suggested that environmental exposure to MeHg is linked to the development of diabetes mellitus (DM). The exact molecular mechanism of MeHg-induced pancreatic β-cell cytotoxicity is still unclear. Here, we found that MeHg (1-4 μM) significantly decreased insulin secretion and cell viability in pancreatic β-cell-derived RIN-m5F cells. A concomitant elevation of mitochondrial-dependent apoptotic events was observed, including decreased mitochondrial membrane potential and increased proapoptotic (*Bax**, Bak*, *p53*)/antiapoptotic (*Bcl-2*) mRNA ratio, cytochrome c release, annexin V-Cy3 binding, caspase-3 activity, and caspase-3/-7/-9 activation. Exposure of RIN-m5F cells to MeHg (2 μM) also induced protein expression of endoplasmic reticulum (ER) stress-related signaling molecules, including C/EBP homologous protein (CHOP), X-box binding protein (XBP-1), and caspase-12. Pretreatment with 4-phenylbutyric acid (4-PBA; an ER stress inhibitor) and specific siRNAs for CHOP and XBP-1 significantly inhibited their expression and caspase-3/-12 activation in MeHg-exposed RIN-mF cells. MeHg could also evoke c-Jun N-terminal kinase (JNK) activation and reactive oxygen species (ROS) generation. Antioxidant *N*-acetylcysteine (NAC; 1mM) or 6-hydroxy-2,5,7,8-tetramethylchroman-2-carboxylic acid (trolox; 100 μM) markedly prevented MeH-induced ROS generation and decreased cell viability in RIN-m5F cells. Furthermore, pretreatment of cells with SP600125 (JNK inhibitor; 10 μM) or NAC (1 mM) or transfection with JNK-specific siRNA obviously attenuated the MeHg-induced JNK phosphorylation, CHOP and XBP-1 protein expression, apoptotic events, and insulin secretion dysfunction. NAC significantly inhibited MeHg-activated JNK signaling, but SP600125 could not effectively reduce MeHg-induced ROS generation. Collectively, these findings demonstrate that the induction of ROS-activated JNK signaling is a crucial mechanism underlying MeHg-induced mitochondria- and ER stress-dependent apoptosis, ultimately leading to β-cell death.

## 1. Introduction

Heavy metals are significant environmental pollutants with hazardous effects on mammal [1]. Many studies have reported that heavy metal-induced deleterious effects are positively correlated with the development of human diseases, including cancer, cardiovascular disease, neurological disorders, and diabetes mellitus (DM) [1,2]. Methylmercury (MeHg), one of the most toxic heavy metals, is a widely distributed, highly poisonous environmental toxicant generated by the methylation of inorganic mercury, one which has become a ubiquitous pollutant causing adverse effects in humans [3]. MeHg can bioaccumulate, magnify, and reach high levels in the ecological food chain, and people can ingest it through food intake, especially fish and seafood, leading to toxicological effects [3,4]. The neurotoxicity of MeHg is well known through worldwide intoxication incidents and experimental studies regarding low concentration exposure [5]. However, growing studies have pointed out a potentially strong association between MeHg and disease development including DM [3,6]. In patients with diagnosed Minamata disease (MeHg poisoning), an increased incidence of DM was observed [7]. Epidemiological studies have also indicated a positive correlation between higher mercury levels in hair/blood and diagnosed DM, particularly in individuals with higher blood levels of mercury [8,9,10]. A study by He et al. [11] reported that exposure to higher levels of mercury in young adults is positively correlated with adverse effects on pancreatic β-cell function, which can increase the risk of developing DM later in life. In animal models, MeHg exposure in rat and mice has been observed to cause pancreatic islet cell damage and apoptosis, resulting in insulin secretion disruption and blood glucose intolerance [12,13,14]. A few studies indicate that MeHg exposure can induce pancreatic β-cell cytotoxicity and dysfunction through a reactive oxygen species (ROS)-dependent pathway [12,15]. However, the action mechanism of MeHg in pancreatic β-cell cytotoxicity is still unclear.

The endoplasmic reticulum (ER), a central intracellular organelle, has essential roles in multiple cellular processes that are required for cell survival and normal functions, including protein synthesis and modification, lipid synthesis, and Ca^2+^ homeostasis and secretion [16]. Disturbances in ER function by some pathophysiological conditions, such as damaged DNA accumulation and toxic chemical exposure, can cause ER stress. Under overwhelming or chronically prolonged ER stress, the normal functions fail to recover, and secretory proteins with unfolded/immature forms are accumulated in the ER, which greatly exceeds its fold capacity, leading to cellular dysfunction and apoptosis [17,18]. ER stress has been demonstrated to be a hallmark of the development of many diseases, such as neurodegenerative diseases, cancer, and DM [19,20]. ER stress and the subsequent unfolded protein response (UPR) have been implicated in β-cell dysfunction and death in type 1 and type 2 DM [21,22]. Due to high insulin demand, β-cells greatly rely on the ER to ensure synthesis and proper folding of pro-insulin, which leads to their being particularly sensitive to ER stress [21]. A growing number of studies have indicated that pancreatic β-cell dysfunction and damage induced by toxic chemicals or pathological stimuli are accompanied by the activation of ER stress signaling [23,24,25,26,27]. Moreover, the previous in vitro studies showed that exposure to mercury compounds can cause morphologic abnormalities in rough ER and display protein degeneration owing to mercury’s affinity for the sulfhydryl groups found on many proteins, leading to ER stress [28,29]. Recently, MeHg has been shown to cause cytotoxicity via an ER stress-mediated apoptotic pathway in various cell types, including neuronal cells, myogenic cells, and embryonic fibroblasts [30,31]. Some toxic metals, such as inorganic arsenic, cadmium, molybdenum, have been found to be capable of inducing an ER stress-dependent signaling pathway to cause pancreatic β-cell dysfunction and death [24,27,32]. However, the role of the ER stress-regulated signaling pathway in MeHg-induced β-cell damage is still unclear.

C-Jun N-terminal kinase (JNK) belongs to the superfamily of MAP-kinases (MAPKs), contributing to cellular events under physiological and pathophysiological conditions [33]. JNK activation can be involved in toxic insult-induced cell dysfunction, apoptosis and death in mammalian cells, including neuronal cells, alveolar epithelial cells, hepatocytes, and pancreatic β-cells [34,35,36]. The JNK signaling pathway has been demonstrated to be involved in cytokine-induced β-cell apoptosis [37,38]. Recent studies have shown that β-cell dysfunction and death induced by toxic chemicals or pathophysiological states are regulated by an oxidative stress-regulated JNK signaling pathway [34,39,40]. Oxidative stress has been reported to be an important risk factor in the onset and progression of β-cell damage and apoptosis, resulting in the development of DM [41,42]. Β-cells are vulnerable targets of ROS attacks, due to their weaker antioxidant defense, leading to induction of β-cell apoptosis and a decline in β-cell function and mass [43]. The JNK pathway has been shown to be involved in oxidative stress-induced pancreatic β-cell cytotoxicity during the diabetic state [44,45]. Exposure to toxic insults such as arsenic, cadmium, palmitate, and tributyltin have been shown to induce oxidative stress and JNK signal activation, causing β-cell apoptosis and death [32,34,39,46]. However, the roles of oxidative stress and JNK signals in MeHg-induced β-cell damage are still unclear.

In this study, we aimed to investigate the crucial roles of ROS and JNK signals in MeHg-induced β-cell dysfunction and death. The involvement of mitochondrial dysfunction and ER stress response in MeHg-induced β-cell apoptosis was also examined and clarified.

## 2. Results

### 2.1. Effects of MeHg on Cell Viability, Apoptosis, and Insulin Secretion in Pancreatic β-Cells

To investigate MeHg-induced pancreatic β-cell damage, we first examined the cytotoxic effect of MeHg on pancreatic β-cell-derived RIN-m5F cells. As shown in Figure 1A, the cell viability of RIN-m5F cells was significantly reduced after treatment with MeHg (1–4 μM) for 24 h in a concentration-dependent manner (1 μM, 90.8 ± 3.5% of control; 2 μM, 54.7 ± 2.8% of control; 3 μM, 17.3 ± 1.1% of control; 4 μM, 7.2 ± 0.5% of control). We next examined whether cell apoptosis was involved in MeHg-induced pancreatic β-cell cytotoxicity. As shown in Figure 1B, the caspase-3 activity (an integral step in apoptosis) was dramatically induced after cells were treated with MeHg (1–3 μM) for 24 h (1 μM, 152.5 ± 8.4% of control; 2 μM, 309.6 ± 13.2% of control; 3 μM, 490.8 ± 21.8% of control). Pretreatment of cells with 5 μM Z-DEVD-FMK (a specific caspase-3 inhibitor) could effectively prevent MeHg-induced increased caspase-3 activity. On the basis of these results, the median effective concentration (EC_50_) of MeHg in RIN-m5F cells was approximately 2 μM, and this concentration was used in subsequent experiments.

Furthermore, we tested the short-term effect of MeHg on insulin secretion from β-cells. As shown in Figure 1C, glucose-stimulated insulin secretion in RIN-m5F cells was significantly and markedly prevented following MeHg treatment for 4 h, which did not change the cell viability (1 μM, 99.6 ± 4.9% of control; 2 μM, 99.2 ± 3.0% of control; 3 μM, 99.3 ± 3.3% of control). These results indicate that MeHg is capable of inducing cytotoxicity, apoptotic activity, and insulin secretion dysfunction in β-cells.

### 2.2. MeHg-Induced Apoptosis Is Mediated by the Mitochondrial-Dependent Pathway in β-Cells

To determine whether MeHg-induced β-cell apoptosis is mediated by a mitochondrial-dependent pathway, we examined the effects of MeHg on MMP and cytochrome *c* release. As shown in Figure 2A, treatment of RIN-m5F cells with MeHg (2 μM) for 8 h markedly induced a depolarization of MMP (64.3 ± 2.8% of control; *p* < 0.05), and a more significant loss was observed after 24 h treatment (18.3 ± 1.1% of control; *p* < 0.05). Treatment with MeHg (2 μM) for 24 h could also effectively increase cytosolic cytochrome c release in RIN-m5F cells (Figure 2B).

Furthermore, the Bcl-2 family plays a crucial role in apoptosis by functioning as inhibitors (Bcl-2) or promoters (Bax and Bak) of cell death. The tumor suppressor p53 acts as an upstream effector of the Bcl-2 family and is also an important activator of the intrinsic apoptosis pathway. We next investigated the effects of MeHg on mRNA expression of Bcl-2, Bax, Bak, and p53. As shown in Figure 2C, a significant decrease in anti-apoptotic *Bcl-2* and increase in pro-apoptotic *Bax**/Bak* and *p53* gene expression levels were observed in RIN-m5F cells treated with 2μM MeHg for 24 h (*p* < 0.05). These results show a significant shift in the pro-apoptotic/anti-apoptotic ratio toward a state associated with apoptosis.

To confirm whether cell apoptosis **i**s involved in MeHg-induced pancreatic β-cell cytotoxicity, we examined phosphatidylserine (PS) externalization (a hallmark of apoptotic events) by annexin V-Cy3/6-CFDA double staining assay and activation of caspase cascade proteases. As shown in Figure 2D, RIN-m5F cells treated with MeHg (2 μM) for 24 h had a markedly increased the number of positive annexin V-Cy3 stained cells (red fluorescence) compared to untreated cells labeled with green fluorescence (6-CFDA). Furthermore, increased levels of protein expression for cleaved forms of caspase-3 and -7 as well as the upstream caspase, caspase-9, were revealed in RIN-m5F cells treated with MeHg at various time intervals (8–24 h) (Figure 2E). These results imply that the mitochondria-dependent apoptosis pathway plays a crucial role in MeHg-induced pancreatic β-cell death.

### 2.3. Involvement of ER Stress Signaling Pathway in MeHg-Induced β-Cell Apoptosis

We next investigated whether ER stress signals are involved in MeHg-induced pancreatic β-cell apoptosis. As shown in Figure 3A, treatment of RIN-m5F cells with 2 μM MeHg effectively induced protein expression of ER stress-related molecules, including CHOP and the spliced form of XBP-1 (XBP-1s), and degradation of full-length caspase-12, a downstream ER stress molecule (55 kDa fragment). However, the protein expression of glucose-regulated protein (GRP)78 and 94 was not changed by MeHg treatment. We then used the specific ER stress inhibitor 4-phenylbutyric acid (4-PBA) and the siRNA-mediated knockdown of CHOP and XBP-1 to examine MeHg-induced apoptotic events. As shown in Figure 3B, pre-treatment of RIN-m5F cells with 4-PBA (2.5 mM) for 1 h prior to MeHg exposure for 24 h effectively prevented the MeHg-induced increased protein expression of CHOP and XBP-1s, degradation of procaspase-12, and caspase-3 and -7 cleavage. In addition, transfection of RIN-m5F cells with both CHOP and XBP-1-specific siRNAs significantly reduced the MeHg-induced increased protein expression of CHOP and XBP-1s, procaspase-12 degradation, and caspase-3 cleavage (Figure 3C,D). These results indicate that exposure to MeHg can induce ER stress response-regulated apoptosis, leading to β-cell death.

### 2.4. JNK Signal Plays an Important Role in MeHg-Induced Pancreatic β-Cell Apoptosis

MAPKs and Akt-mediated pathways have been reported to play a role in cell apoptosis [47]. We examined whether activation of MAPKs and Akt was involved in MeH-induced β-cell apoptosis. As shown in Figure 4, the protein expression of phosphorylated JNK, ERK1/2, and Akt was significantly increased in RIN-m5F cells treated with MeHg (2 μM) for 30–120 min. However, MeHg treatment did not affect p38 protein phosphorylation in RIN-m5F cells. Pretreatment of cells with the pharmacological JNK inhibitor (SP600125), but not ERK1/2 inhibitor (PD98059) and Akt inhibitor (LY 294002), for 1 h prior to MeHg exposure effectively prevented the increase in caspase-3 activity (Figure 5A) and the reduction in cell viability (Figure 5B). Moreover, the JNK inhibitor (SP600125) potently reversed the MeHg-induced JNK phosphorylation (Figure 5C), MMP loss (Figure 5D), protein expression of CHOP, XBP-1s, and cleaved caspase-3, -7, and -12, (Figure 5E), and glucose-stimulated insulin secretion inhibition (Figure 5F) in RIN-m5F cells. Transfection of RIN-m5F cells with JNK-specific siRNA effectively attenuated MeHg-induced caspase-3 activity (Figure 6A), MMP loss (Figure 6B), and protein expression of CHOP, XBP-1s, cleaved caspase-3 and -12 (Figure 6C), and phosphorylated JNK1/2 (Figure 6D). These results imply that the JNK-activated downstream-mediated mitochondria-dependent and ER stress-regulated apoptosis pathway contributes to MeHg-induced pancreatic β-cell death.

### 2.5. Intracellular ROS Production Is an Important Event in MeHg-Induced Pancreatic β-Cell Injury

We next examined the role of ROS in MeHg-induced β-cell injury. As shown in Figure 7A, flow cytometric analysis showed that exposure of cells to 2 μM MeHg for 0.5-4 h triggered a significant increase in intracellular ROS generation in a time-dependent manner. This ROS generation, as well as decreased cell survival, could be effectively reversed by 6-hydroxy-2,5,7,8-tetramethylchroman-2-carboxylic acid (Trolox, 100 μM) and antioxidant *N*-acetylcysteine (NAC, 1 mM) (Figure 1A and Figure 7A), indicating that ROS plays an important regulating role in MeHg-induced β-cell injury. In addition, pretreatment of RIN-m5F cells with NAC (1 mM) for 1 h prior to MeHg (2 μM) exposure effectively diminished the mitochondrial dysfunction (including MMP loss, cytosolic cytochrome c release, and mRNA expression of Bcl-2, Bax, Bak, and p53) (Figure 2A–C), apoptotic responses (including annexin V-Cy3 staining for PS externalization and cleaved caspase-3/-7/-9 protein expression) (Figure 2D,2E), ER stress responses (including protein expression of CHOP and XBP-1s and procaspase-12 degradation) (Figure 7B), JNK phosphorylation (Figure 7C), and glucose-stimulated insulin secretion inhibition (Figure 5F). However, pretreatment of RIN-mF cells with the specific JNK inhibitor (SP600125) did not affect MeHg-induced ROS production (Figure 7D). These results point out that intracellular ROS generation may contribute to the activation of the JNK signal downstream-regulated mitochondria-dependent and ER stress-mediated apoptotic pathway, leading to pancreatic β-cell injury by MeHg.

## 3. Discussion

DM, one of the fastest growing chronic diseases in the USA, the EU, and other countries, is a metabolic disease characterized by hyperglycemia due to defects in insulin secretion, insulin action, or both, leading to hyperglycemia [48]. Dysfunction of pancreatic islet β-cells, which are the main parts involved in producing and secreting insulin, plays a critical role in the pathogenesis of type 1 and type 2 DM [49]. Several metals, including MeHg, have been reported to be epidemiologically linked to the incidence of DM [9,11,50]. MeHg is known to be the most poisonous among the mercury compounds and a global pollutant ubiquitously present in the environment that can induce severe toxic effects in humans [3]. Prenatal exposure to MeHg from maternal ingestion of contaminated seafood with a MeHg concentration of approximately 2 mg/kg was associated with an increased risk of neurodevelopmental deficits and neurotoxic anomalies [51]. Li et al. [52] indicated that a MeHg the concentration up to 8–174 μg/kg (approximately 0.4–0.81 μM) can be detected in rice and 10–590 μg/kg (approximately 0.5–2.74 μM) in fish from mercury polluted areas. An epidemiological study conducted in Mexico showed that mercury levels in serum from patients with type 2 DM in Mexico were 1.12 ± 0.97 μg-Hg/L, which were considerably higher compared to healthy age-matched subjects [53]. Nakagawa (1995) has reported that mercury levels in hair from people with diseases (including DM) were 2.08–36.5 mg-Hg/L [54]. A prospective cohort study of young adults in the USA indicated that higher mercury exposure at baseline was significantly associated with a higher incidence of DM and decreased homeostasis in an assessment of β-cell function index, suggesting that people with high mercury exposure in young adulthood may have an elevated risk of developing diabetes later in life [11]. Skalnaya and Demidov [10] reported that women diagnosed with DM were found to have increased hair mercury levels in their hair (1.21 mg-Hg/kg) compared to healthy subjects (0.62 mg-Hg/kg). Chang et al. [8] further observed a statistically significant and positive association between higher blood mercury levels (0.5–86.8 μg/L) and defective pancreatic β-cell function. Furthermore, a growing number of studies are reporting an association between MeHg exposure and the risk of developing DM [6,55]. An increased incidence of DM has been reported in patients with diagnosed Minamata disease [7,56]. Disturbance and damage to pancreatic islet cells resulting in hyperglycemia have been shown in postmortem examinations of cases of Minamata disease and in in vivo models [7,12,13,14]. A few studies suggest that MeHg can cause pancreatic β-cell dysfunction and damage through an ROS-mediated signaling pathway [12,15]. However, the action mechanism of MeHg on pancreatic β-cell injury is still unclear. In the present study, the major findings were that exposure of RIN-m5F cells to low-concentration MeHg led to insulin secretion dysfunction and induced mitochondria- and ER stress-dependent apoptosis via a ROS-mediated JNK signaling pathway, resulting in β-cell death.

The ER is a cellular compartment responsible for cell survival and many important cellular functions including the biosynthesis, folding, and secretion of newly synthesized proteins as insulin [57]. In healthy β-cells, proinsulin is quite susceptible to misfolding (estimated at around 20% misfolds); therefore, a well-developed ER is an important to recognize and remove misfolded proinsulin [58]. However, many pathophysiological factors can perturb ER function, which can prevent regulation of ER homeostasis and stress. Overactivation of the UPR by ER stress elicits a signaling cascade, which destroys/fails to maintain ER homeostasis, leading to β-cell dysfunction/death and the pathogenesis of DM [18,21,59]. Excessive ER stress has been shown to contribute to cell death in pancreatic β-cells of DM patients [60,61]. Under excessive or unresolved ER stress, the persistent stimulation of the UPR switches from an adaptive role to an apoptotic role, triggering apoptosis via activation of the proapoptotic UPR effector CHOP [62,63]. CHOP, which is also known as growth arrest and DNA damage-inducible gene 153 (GADD153), is identified as part of the CCAAT enhancer binding protein (C/EBP) family, which is present at low constitutive levels in the cytosol, but exhibits markedly increased expression in response to ER stress-mediated apoptosis [17,63]. Growing evidence has shown that CHOP expression is implicated in pancreatic β-cell pathophysiological processes in the development of type 1 and type 2 DM [19,64]. The response of CHOP-associated apoptosis compromising the overall functionality of pancreatic β-cells in the diabetic condition can be abrogated in CHOP-deficient mice and cells compared with normal subjects [65]. Disruption of the *Chop* gene ameliorated β-cell loss and DM in the Akita mouse [17,19]. In addition, XBP-1 is a required transcription factor for cellular responses to ER stress. Upon sensing ER stress, the spliced form of XBP-1 (XBP-1s, an active transcription factor) is produced and activated in the UPR [66]. XBP-1 plays a distinct inhibitory role in β-cells, which is associated with DM development [67]. Marchetti et al. [61] showed that exposure to high-glucose conditions could induce *XBP-1* gene expression in type 2 diabetic islets, but not in control islets. Allagnat and colleagues also reported that prolonged XBP1s overproduction hampered β-cell function via impairment of insulin secretion function and inhibition of *Mafa*, *Pdx1*, and *Insulin* gene expression, eventually leading to Bax translocation, cytochrome c release, caspase-3 activation, and β-cell apoptosis [68]. In experimental paradigms, exposure to toxic chemicals has been shown to induce ER stress-triggered apoptosis in mammalian cells, such as neuronal cells, lung epithelial cells, pancreatic islets, and β-cells, which is mainly accompanied by CHOP and XBP-1 activation [23,25,35,69,70]. Pretreatment with 4-PBA (an ER stress inhibitor) and/or transfection of cells with CHOP- and XBP-1-specific siRNAs has been shown to significantly reduce the levels of CHOP and XBP-1s expression and prevent apoptosis responses in β-cells treated with cadmium- or 4-methyl-2,4-bis(4-hydroxyphenyl)pent-1-ene (MBP) [24,25], suggesting that CHOP/XBP-1 may play important roles in β-cell apoptosis. However, only a few studies have indicated that the involvement of a CHOP and/or XBP-1 signal activation-mediated apoptosis pathway in MeHg exposure-induced neuronal cell death [30,71], but not in β-cells. The role of CHOP/XBP-1 activation in MeHg-induced β-cell apoptosis is still unclear. In the present study, we found that MeHg was capable of destroying mitochondrial function, including loss of MMP, increased cytosolic cytochrome c release, and changes in pro-apoptotic (*Bax*, *Bak*, and *p53*)/ anti-apoptotic (*Bcl-2*) gene expression, leading to β-cell apoptosis. Furthermore, exposure of β-cells to MeHg significantly increased the expression of CHOP and XBP-1s and degradation of procaspase-12. Pretreatment of RIN-m5F cells with 4-PBA and specific siRNA transfection for CHOP- and XBP-1 dramatically abrogated MeHg-induced CHOP/XBP-1s protein expression and caspase-3 activation. These results indicate that MeHg exposure causes β-cell death through mitochondrial- and ER stress-dependent apoptotic pathways.

Excessive ROS generation has various deleterious effects on mammalian cells contributing to the development of many diseases including DM [41,59]. In the diabetic state, exposure to high levels of glucose induces β-cell damage and apoptosis along with ROS overproduction [72]. Furthermore, toxic heavy metals-induced oxidative stress is positively associated with the development of human diseases such as neurodegenerative diseases, cancer, and DM [2,50]. A growing number of studies have shown that toxic metals such as arsenic, cadmium, copper, tributyltin can induce oxidative stress, consequently causing β-cell damage and apoptosis in vitro and in vivo [32,34,39,40]. On the other hand, JNK signaling, which is activated by various stress factors, such as pathological responses and environmental chemicals, is well documented to be involved in the regulation of a wide range of cellular processes, including cell proliferation, differentiation, migration, and apoptosis [33,73]. JNK functions have been reported as a pro-apoptotic kinase by translocating Bax in the mitochondria and an anti-apoptotic activity by inhibition Bcl-2, that plays an important role in apoptosis induced by intrinsic as well as extrinsic pathway [33]. Activation of the JNK pathway has been shown to be induced by arsenic or cadmium, leading to apoptosis in neuronal cells [35,74]. Recent studies have indicated that activations of the JNK, CHOP, or XBP-1 signal-triggered apoptotic pathway is involved in toxic chemical-induced β-cell damage and death [24,25,27,75,76]. Moreover, several studies have highlighted that toxic metals such as arsenic, cadmium, Cu^2+^/PDTC complex, Tributyltin can induce oxidative stress, followed by the downstream regulation of JNK activation and/or mitochondrial dysfunction resulting in β-cell apoptosis, and these responses can be reversed by the antioxidant NAC [32,34,39,40]. The studies of Chen et al. [12,13,15] have reported that exposure to MeHg can significantly induce oxidative stress, causing pancreatic islet β-cell dysfunction and apoptosis in vitro and in vivo. However, there are no studies that fully elucidate the precise mechanism and the relationship between oxidative stress and JNK signaling underlying MeHg-induced β-cell damage. In this study, our results, for the first time, demonstrated MeHg significantly increased ROS generation and the phosphorylated levels of JNK protein in RIN-m5F cells. Pretreatment with antioxidant NAC and SP600125 (JNK inhibitor) and transfection with JNK-specific siRNA effectively attenuated the MeHg-induced apoptotic events, MMP loss, CHOP/XBP-1 activation, and insulin secretion dysfunction. The increased ROS production in MeHg-treated RIN-m5F cells was effectively prevented by NAC, but not SP600125. These results suggest that **the** oxidative stress-activated JNK signaling pathway plays a crucial role in the downstream regulation of both mitochondria- and ER stress-dependent apoptosis in MeHg-induced pancreatic β-cell death.

## 4. Materials and Methods

### 4.1. Materials

Unless otherwise specified, all chemicals and laboratory plastic wares were purchased from Sigma-Aldrich (St. Louis, MO, USA) and Falcon Labware (Becton, Dickinson and Company, Franklin Lakes, NJ, USA), respectively. RPMI 1640 medium, fetal bovine serum (FBS), and antibiotics were purchased from Gibco/Invitrogen (Thermo Fisher Scientific Inc., Waltham, MA, USA). Mouse or rabbit monoclonal antibodies specific for caspase-3 (Cat. No.: #9661), caspase-7 (Cat. No.: #9491), caspase-9 (Cat. No.: #9508), cytochrome c (Cat. No.: #11940), GRP 78 (Cat. No.: #3183), GRP 94 (Cat. No.: #2104), CHOP (Cat. No.: #2895), XBP-1 (Cat. No.: #40435), phosphorylated (p)-ERK1/2 (Cat. No.: #4377), p-p38 (Cat. No.: #9216), p-Akt (Cat. No.: #9271), ERK1/2 (Cat. No.: #9102), p38 (Cat. No.: #8690), Akt (Cat. No.: #9272), and β-actin (Cat. No.: #8457), and secondary antibodies [horseradish peroxidase (HRP)-conjugated anti-mouse IgG (Cat. No.: #7076) or anti-rabbit IgG (Cat. No.: #7074)] were purchased from Cell Signaling Technology (Danvers, MA, USA)**.** The antibody specific for caspase-12 (Cat. No.: sc-21747) was purchased from Santa Cruz Biotechnology (Santa Cruz, CA, USA). Antibodies specific for p-JNK (Cat. No.: AP0473) and JNK-1 (Cat. No.: A5051) were purchased from ABclonal (Woburn, MA, USA).

### 4.2. Cell Culture

A rat pancreatic islet β-cell line, RIN-m5F, which is capable of producing and secreting insulin, was purchased from American Type Culture Collection (ATCC; CRL-11605). Cells were cultured in a humidified chamber with a 5 CO_2_–95% air mixture at 37 °C and maintained in RPMI 1640 medium supplemented with 10% FBS and antibiotics (100 U/mL of penicillin and 100 μg/mL of streptomycin).

### 4.3. Cell Viability Assay

Cells were washed with fresh medium and cultured in 96-well plates (2 × 10^4^ cells/well) and then stimulated with MeHg (1–4 μM) or not in the absence or presence of NAC (1 mM), trolox (100 μM), SP600125 (10 μM), PD98059 (10 μM), or LY294002 (5 μM) for 24 h. After incubation, **the** medium was aspirated and cells were incubated with fresh medium containing 0.2 mg/mL 3-(4,5-dimethyl thiazol-2-yl-)-2,5-diphenyl tetrazolium bromide (MTT). After 4 h, the medium was removed and blue formazan crystals were dissolved in 100 μL of dimethyl sulfoxide (DMSO). Absorbance at 570 nm was measured using a Bio-Tek uQuant Microplate Reader (MTX Lab Systems, Winooski, VT, USA).

### 4.4. Determination of Caspase-3 Activity

Caspase-3 activity was assessed using a Caspase-3 Activity Assay Kit (Cell Signaling Technology, Inc., Danvers, MA, USA). RIN-m5F cells were seeded at 2 ×10^5^ cells/well in a 24-well plate and treated with MeHg (1–3 μM) or not in the absence or presence of Z-DEVD-FMK (5 μM), SP600125 (10 μM), PD98059 (10 μM), or LY294002 (5 μM) at 37 °C. At the end of treatment (24 h), the cell lysates were incubated at 37 °C with 10 μM Ac-DEVD-AMC, a caspase-3/CPP32 substrate, for 1 h. The fluorescence of the cleaved substrate was measured using a spectrofluorometer (Gemini XPS Microplate Reader, Molecular Devices, San Jose, CA, USA) at an excitation wavelength of 380 nm and an emission wavelength of 460 nm.

### 4.5. Determination of Insulin Secretion

To measure the amount of insulin secretion in RIN-m5F cells after being exposed to MeHg (1–3 μM) or not, in the absence or presence of NAC (1 mM) or SP600125 (10 μM), cells were incubated in Krebs-Ringer buffer (KRB) with 2.8 and 20 mM glucose for 1 h to stimulate insulin secretion, and the supernatant was immediately collected and stored at −20 °C [25,27]. To measure the amount of insulin secretion, aliquots of samples were assayed using an insulin antiserum immunoassay kit (Mercodia, Uppsala, Sweden) according to the manufacturer’s instructions.

### 4.6. Detection of Apoptotic Cells

Apoptotic cells were detected using an Annexin V-Cy3^TM^ apoptosis detection kit (Sigma-Aldrich, St. Louis, MO, USA) as previously described by Chang et al. [34]. Briefly, cells were treated with MeHg (2 μM) or not in the absence or presence of NAC (1 mM) for 24 h. After treatment, cells were washed twice with PBS (PH 7.4), and the incubated with Annexin V-Cy3 (Ann Cy3) and 6-carboxy fluorescein diacetate (6-CFDA) simultaneously. After being labeled at room temperature, cells were immediately observed by fluorescence microscopy (Axiovert 200, Zeiss, 200×). Ann Cy3 was available for binding to phosphatidylserine, which was observed as red fluorescence. In addition, cell viability was detected by 6-CFDA, which was hydrolyzed to 6-CF and appeared as green fluorescence. Cells in the early stage of apoptosis were labeled with both Ann Cy3 (red) and 6-CF (green).

### 4.7. Measurement of ROS Production

Intracellular ROS generation was monitored by flow cytometry using the sensitive fluorescent probe, 2′, 7′-dichlorofluorescin diacetate (DCFH-DA), to measure intracellular hydroxyl, hydrogen peroxide, and other ROS activity (Sigma-Aldrich, St. Louis, MO, USA), and dihydroethidium (DHE), which is highly selective for O_2_^•−^ (Invitrogen Molecular Probes, Inc, Eugene, OR, USA). In brief, RIN-m5F cells were seeded at 2 × 10^5^ cells/well in a 24-well plate and incubated with MeHg for 0.5–4 h in the absence or presence of NAC (1 mM), trolox (100 μM), or SP600125 (10 μM) (1 h pre-treatment). At the end of various treatment durations, cells were incubated with medium containing 20 μM DCFH-DA or DHE for 15 min at 37 °C. After incubation with the dye, cells were harvested and washed twice with phosphate-buffered saline (PBS), then re-suspended in ice-cold PBS and placed on ice in a dark environment. The intracellular peroxide levels were measured by a flow cytometer (FACScalibur, Becton, Dickinson and Company, San Jose, CA, USA), using CellQuest software version 5.1 (Becton, Dickinson and Company, San Jose, CA, USA).

### 4.8. Detection of Mitochondrial Membrane Potential (MMP)

MMP was analyzed using a fluorescent probe, DiOC_6_, which is a positively charged mitochondria-specific fluorophore. Briefly, RIN-m5F cells were seeded at 2 × 10^5^ cells/well in a 24-well plate and incubated with or without MeHg for 8 and 24 h in the absence or presence of NAC (1 mM) or SP600125 (10 μM) (1 h pre-treatment). At the end of treatment, cells were incubated with medium containing 100 nM 3,3′-dihexyloxacarbocyanine iodide (DiOC_6_) for 30 min at 37 °C. After incubation with the dye, cells were harvested and washed twice with PBS, and resuspended in ice-cold PBS. MMP was analyzed by a flow cytometer (FACScalibur, Becton, Dickinson and Company, San Jose, CA, USA), using CellQuest software version 5.1 (Becton, Dickinson and Company, San Jose, CA, USA).

### 4.9. Real-Time Quantitative Reverse-Transcription Polymerase Chain Reaction (RT-qRT-PCR) Analysis

The expression of apoptosis-related genes was evaluated by real-time quantitative RT-PCR analysis, as previously described [40]. Briefly, intracellular total RNA was extracted using RNeasy kits (Qiagen, Hilden, Germany) and reverse transcribed into cDNA using AMV RTase, a reverse transcriptase enzyme (Promega Corporation, Madison, WI, USA), according to the manufacturer’s instructions. Each sample (2 μL cDNA) was then tested with Real-Time SYBR Green PCR reagent (Invitrogen, Carlsbad, CA, USA) with rat specific primers (as follows: Bcl-2, forward: 5′-CTTTGTGGAACTGTACGGCCCCAGCATGCG-3′ and reverse: 5′-ACAGCCTGCAGCTTTGTTTCATG-GTACATC-3′; Bax, forward: 5′-GGGAATTCTGGAGCTGCAGAGGATGATT-3′ and reverse: 5′-GCGGATCCAAGTTGC-CATCAGCAAACAT-3′; Bak, forward: 5′-TTTGGCTACCGTCTGGCC-3′ and reverse: 5′-GGCCCAACAGAACCACACC-3′; p53, forward: 5′- GGACGACAGGCAGACTTTTC-3′ and reverse: 5′-CAGATGCAGGGCGGTATTTT-3′; and β-actin, forward: 5′-ATTGTAACCAACTGGGACG-3′ and reverse: 5′-TCTCCAGGGAGGAAGAGG-3′) in a 25 μL reaction volume, and amplification was performed using an ABI StepOnePlus™ Sequence Detection System (Applied Biosystems, Thermo Fisher Scientific, Inc., Waltham, MA, USA). Cycling conditions were 10 min of polymerase activation at 95 °C followed by 40 cycles at 95 °C for 15 s and 60 °C for 60 s. Real-time fluorescence detection was performed during the 60 °C annealing/extension step of each cycle. Melt-curve analysis was performed on each primer set to ensure that no primer dimers or nonspecific amplifications were present under the optimized cycling conditions. After 40 cycles, data analysis was performed using StepOne^TM^ software version 2.1 (Applied Biosystems, Thermo Fisher Scientific, Inc., Waltham, MA, USA). All amplification curves were analyzed with a normalized reporter (R_n_, ratio of fluorescence emission intensity to fluorescence signal of passive reference dye) with a threshold of 0.2 to obtain the C_T_ values (threshold cycle). The reference control genes were measured with four replicates in each PCR run, and their average C_T_ was used for relative quantification analysis [77]. The expression data were normalized by subtracting the mean value of reference gene C_T_ from the C_T_ value (ΔC_T_). The fold change value was calculated by the 2^-ΔΔC^_T_ method, where ΔΔC_T_ represents ΔC_T-condition of interest_–ΔC_T-control_. Prior to conducting statistical analysis, the fold change from the mean of the control group was calculated for each individual sample.

### 4.10. Small Interference RNA Transfection

Specific small interference-RNA (si-RNA) against CHOP (Dharmacon, Inc., Lafayette, CO, USA), XBP-1 (Thermo Fisher Scientific Inc., Waltham, MA, USA), JNK, and control siRNA (Cell Signaling Technology) were used for transfection. RIN-m5F cells were seeded in 6-well culture plates and transfected with si-RNA using Lipofectamine RNAi MAX (Gibco/Invitrogen, Life Technologies, Waltham, MA, USA) according to the manufacturer’s instructions. Cellular levels of the proteins specific for si-RNA transfection were checked by Western blot, and all experiments were performed at 24 h after transfection.

### 4.11. Western Blot Analysis

RIN-m5F cells were seeded at 1 × 10^6^ cells/well in a 6-well culture plate treated with MeHg or not in the absence or presence of NAC (1 mM), 4-PBA (2.5 mM), SP600125 (10 μM), or the specific siRNA transfection for CHOP, XBP-1, and JNK. At the end of various treatment durations, protein expression levels were analyzed by Western blot as previously described by Huang et al. [24,25]. In brief, equal amounts of protein (50 μg per lane) were subjected to electrophoresis on 10% (*w*/*v*) SDS-polyacrylamide gels and transferred to polyvinylidene difluoride (PVDF) membranes. The membranes were blocked for 1 h in PBST (PBS, 0.05% Tween-20) containing 5% nonfat dry milk. After blocking, the membranes were incubated with mouse or rabbit monoclonal antibodies specific against caspase-3, caspase-7, caspase-9, caspase-12, GRP 78, GRP 94, CHOP, XBP-1, p-JNK, p-ERK1/2, p-p38, p-Akt, JNK-1, ERK1/2, p38, Akt, and β-actin in 0.1% PBST (1:1000) for 12-16 h at 4 °C. After 3 additional washes in 0.1% PBST (15 min each), the respective HRP-conjugated secondary antibodies were applied (1:2500 in 0.1% PBST) for 1 h at 4 °C. The antibody-reactive bands were detected by enhanced chemiluminescence reagents (Pierce^TM^, Thermo Fisher Scientific Inc., Waltham, MA, USA) and analyzed by a luminescent image analyzer (ImageQuant™ LAS-4000; GE Healthcare Bio-Sciences Corp., Piscataway, NJ, USA). For cytosol cytochrome *c* expression, the detection was performed as previously described by Chang et al. [34]. In brief, at the end of treatments, cells were detached, washed twice with PBS, and then homogenized with a mortar and pestle in the extract buffer (0.4 M mannitol, 25 mM MOPS (pH 7.8), 1 mM EGTA, 8 mM cysteine, and 0.1% (*w*/*v*) bovine serum albumin). The cell debris was removed via centrifugation at 6000× *g* for 2 min. The supernatant was recentrifuged at 12,000× *g* for 15 min. The supernatant (cytosolic fraction) was used to detect cytochrome c expression by Western blot analysis.

### 4.12. Statistical Analysis

Data are presented as mean ± standard deviation (SD) of at least 4 independent experiments. All data analyses were performed using SPSS software version 12.0 (SPSS, Inc., Chicago, IL, USA). For each experimental condition, significant differences were assessed by one-way analysis of variance (ANOVA) followed by Tukey’s post hoc test; p value <0.05 was considered significant.

## 5. Conclusions

In conclusion, this study elucidates that MeHg induces pancreatic β-cell cytotoxicity through the mitochondria- and ER stress-dependent apoptotic pathways. More importantly, oxidative stress-activated JNK signaling is identified as a crucial mechanism underlying MeHg-induced β-cell apoptosis. These observations provide further evidence to support the supposition that MeHg is an environmental risk factor for the development of DM.

## Figures and Tables

**Figure 1 ijms-23-02858-f001:**
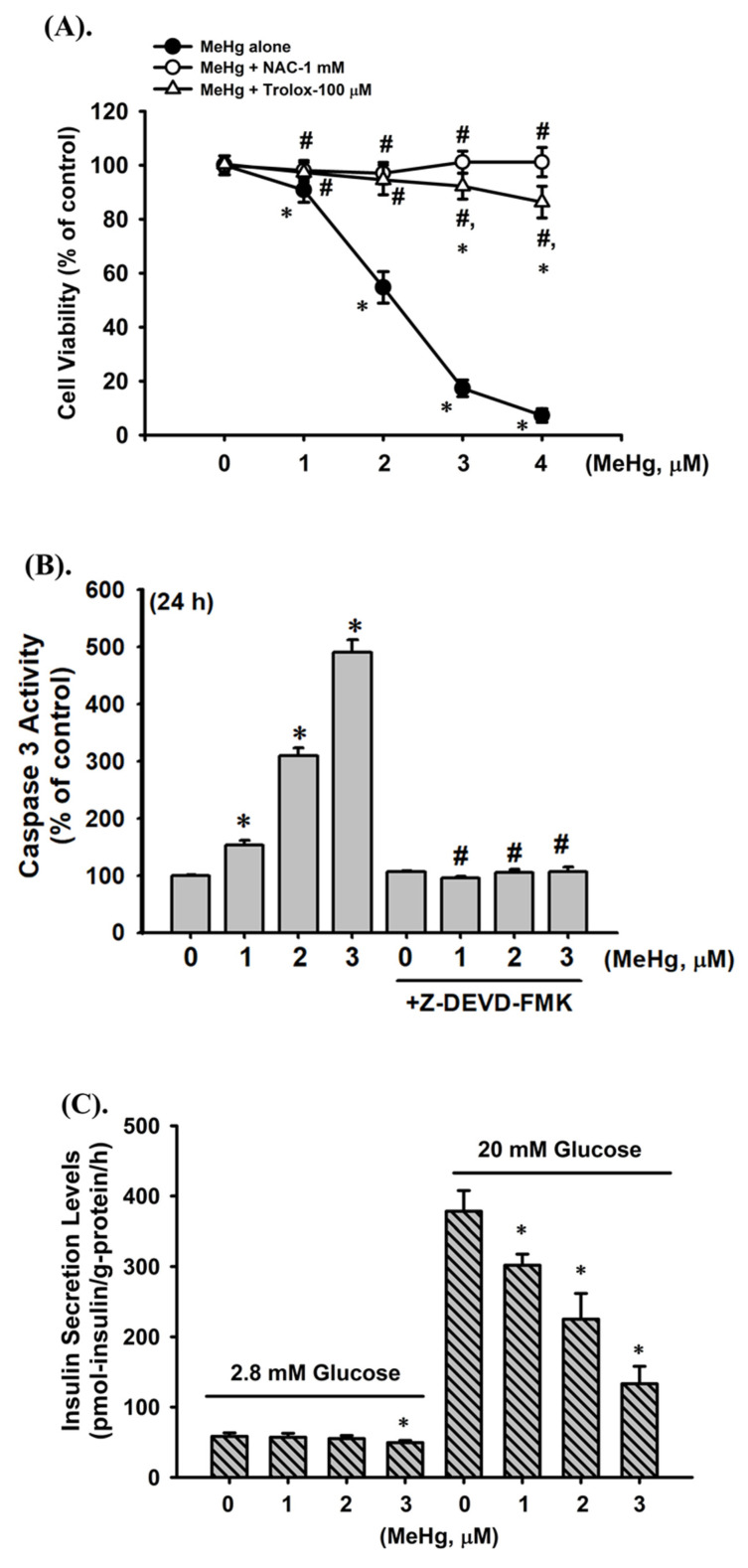
Effects of MeHg on cytotoxicity and insulin secretion in RIN-m5F cells. (**A**) Cells were treated with MeHg (1–4 μM) for 24 h in absence or presence of NAC (1 mM) and trolox (100 μM) for 1 h prior to addition of MeHg. Cell viability was detected using MTT assay. (**B**) RIN-m5F cells were treated with MeHg (1–3 μM) for 24 h in absence or presence of 5 μM Z-DEVD-FMK (a caspase-3 inhibitor) for 1 h prior to addition of MeHg. Caspase-3 activity was detected using a Caspase-3 Activity Assay Kit. (**C**) Cells were treated with MeHg (1–3 μM) or not for 4 h. Low/high glucose-stimulated insulin secretion was measured by immunoassay kit. Data are presented as mean ± SD of six independent experiments assayed in triplicate. * *p* < 0.05 compared to vehicle control; **^#^**
*p* < 0.05 compared to MeHg treatment alone.

**Figure 2 ijms-23-02858-f002:**
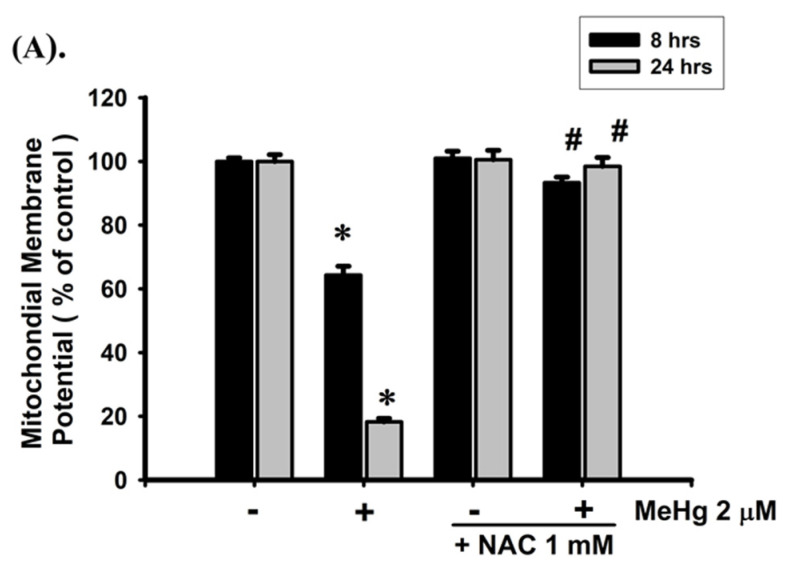
MeHg induced mitochondrial dysfunction and apoptosis in RIN-m5F cells. Cells were treated with MeHg (2 μM) for 8–24 h in absence or presence of NAC (1 mM) for 1 h prior to addition of MeHg. (**A**) Depolarization of mitochondrial membrane potential (MMP) was detected using flow cytometry. (**B**) Release of cytosolic cytochrome c was examined using Western blot analysis. (**C**) Expression of anti-apoptotic (*Bcl-2*) and pro-apoptotic (*Bax*, *Bak* and *p53*) genes was analyzed by real-time quantitative RT-PCR. (**D**) Apoptotic cells stained with Annexin V-Cy3 (red) and 6-CFDA (green) fluorescent probe were observed using fluorescence microscopy (magnification, 200×; scale bar: 100 μm) (a, control; b, MeHg-2 μM; c, NAC-1 mM; d, MeHg-2 μM+NAC-1 mM). (**E**) Levels of protein expression for cleaved caspase-3, -7, and -9 were examined using Western blot analysis. Data in A and C are presented as mean ± SD of six independent experiments assayed in triplicate. * *p* < 0.05 compared to vehicle control; ^#^
*p* < 0.05 compared to MeHg treatment alone. Results shown in B and E are representative images of at least three independent experiments. β-actin was used as loading control.

**Figure 3 ijms-23-02858-f003:**
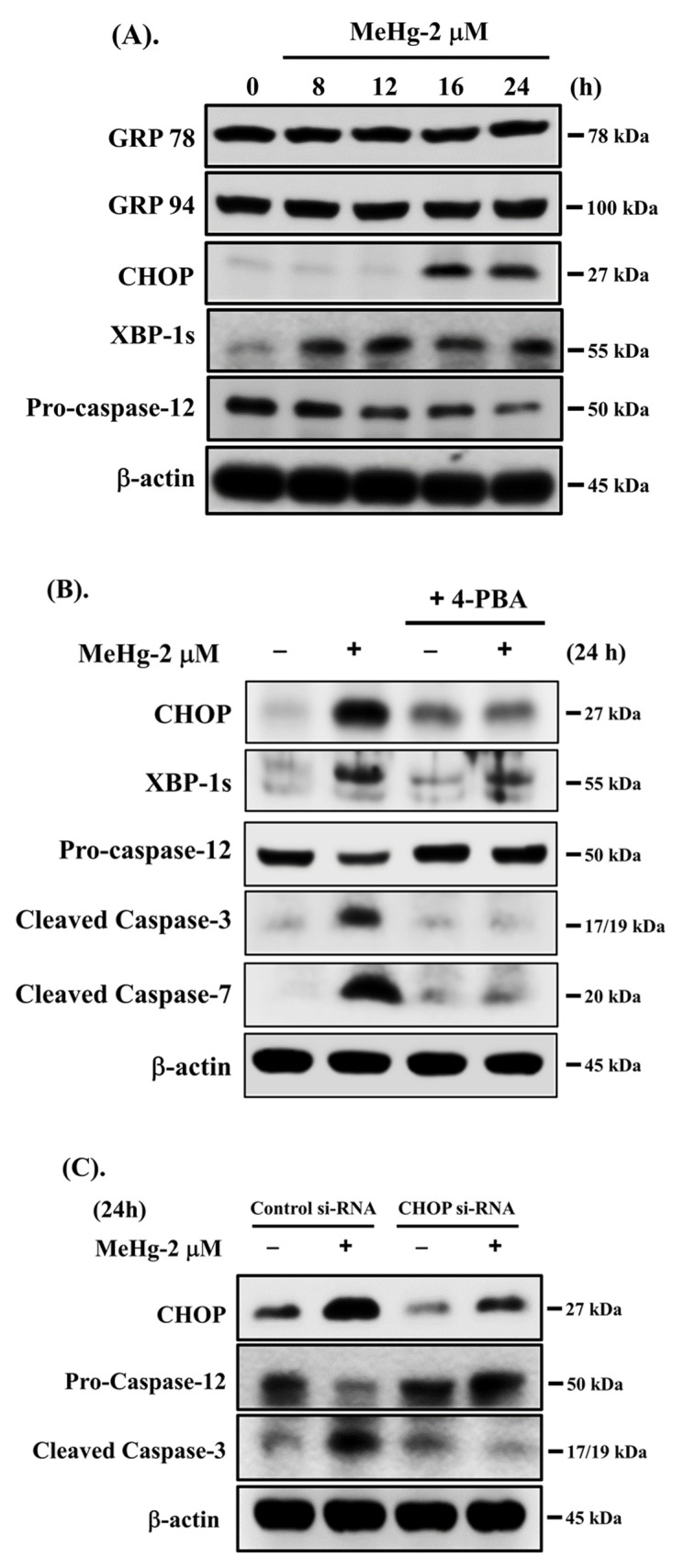
Involvement of ER stress-regulated signaling in MeHg-induced β-cell apoptosis. (**A**) RIN-m5F cells were treated with MeHg (2 μM) for different time intervals, and protein expression of GRP78, GRP94, CHOP, XBP-1s, and procaspase-12 was analyzed by Western blot analysis. (**B**) RIN-m5F cells were treated with MeHg (2 μM) for 24 h in the absence or presence of ER stress inhibitor 4-phenylbutyric acid (4-PBA; 2.5 mM) for 1 h prior to exposure to MeHg. Protein expression of CHOP, XBP-1s, caspase-3 and -7, and procaspase-12 was examined using Western blot analysis. In addition, RIN-m5F cells were transfected with control siRNA or siRNA specific for (**C**) CHOP and (**D**) XBP-1 prior to treatment with MeHg (2 μM) for 24 h. Protein expression of CHOP, XBP-1, caspase-3, and procaspase-12 was examined using Western blot analysis. Results shown are representative images of at least three independent experiments. β-actin was used as loading control.

**Figure 4 ijms-23-02858-f004:**
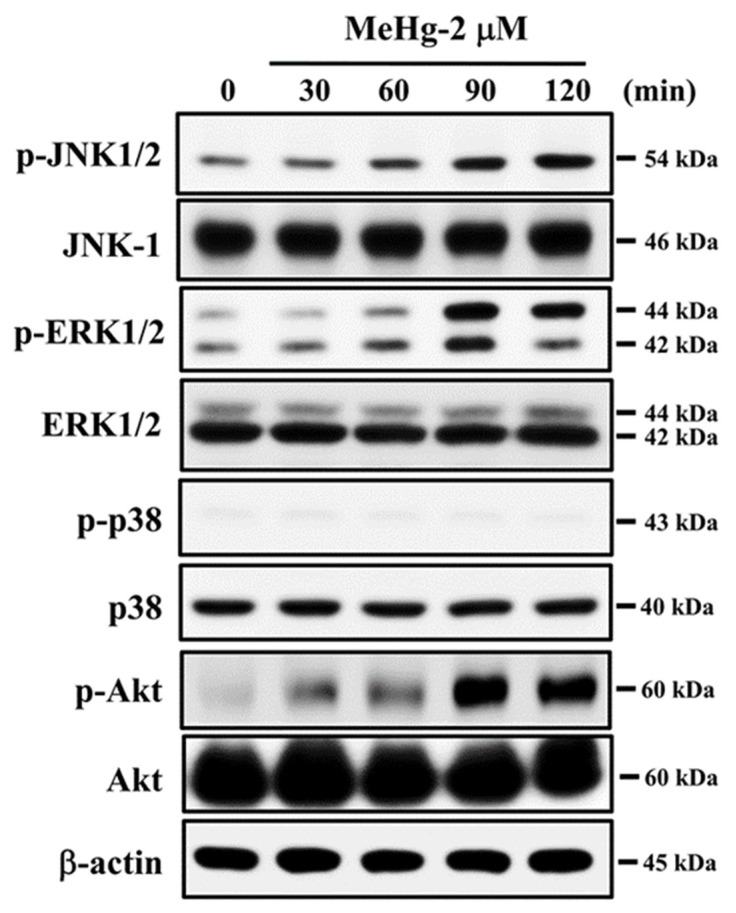
Effects of MeHg on phosphorylation of MAPKs and Akt in β-cells. RIN-m5F cells were treated with MeHg (2 μM) for 30–120 min. Levels of protein phosphorylation for JNK1/2, ERK1/2, p38, and Akt were examined using Western blot analysis. Results shown are representative images of at least three independent experiments. β-actin was used as loading control.

**Figure 5 ijms-23-02858-f005:**
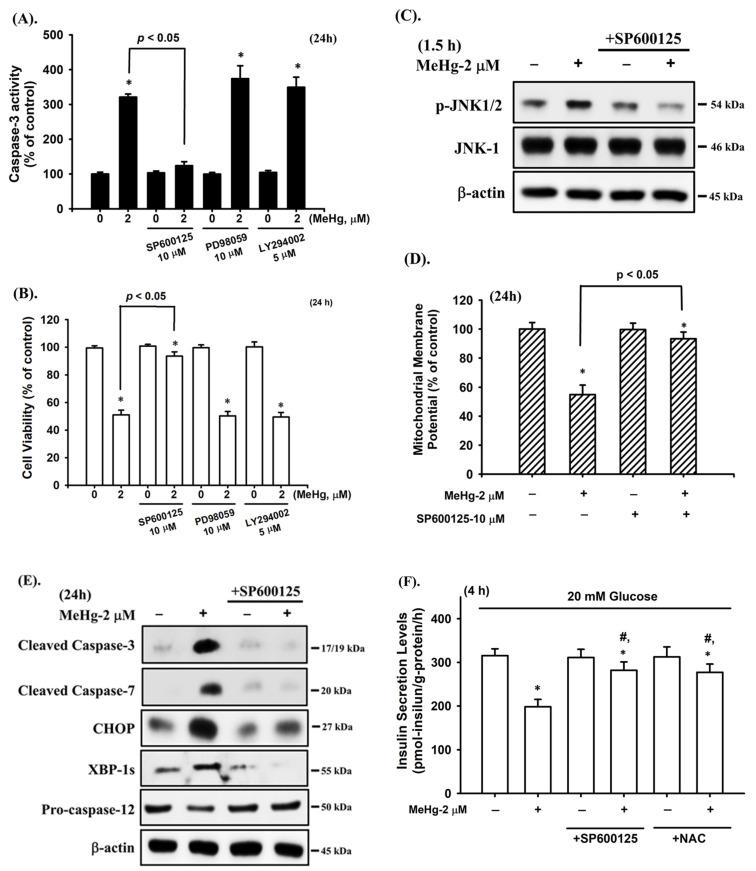
JNK-mediated signaling plays a critical role in MeHg-induced β-cell apoptosis. RIN-m5F cells were treated with MeHg (2 μM) for different time intervals in the absence or presence of specific inhibitors of JNK (SP600125; 10 μM), ERK1/2 (PD98059; 10 μM), and Akt (LY294002; 5 μM) for 1 h prior to the addition of MeHg. (**A**) Caspase-3 activity was measured using a Caspase-3 Activity Assay Kit. (**B**) Cell viability was detected using MTT assay. (**C**) Phosphorylation levels of JNK1/2 protein were examined using Western blot analysis. (**D**) Mitochondrial membrane potential (MMP) depolarization was detected using flow cytometry. (**E**) Protein expression of caspase-3 and -7, procaspase-12, CHOP, and XBP-1s was examined using Western blot analysis. (**F**) RIN-m5F cells were pretreated with JNK inhibitor (SP600125; 10 μM) and antioxidant NAC (1 mM) for 1 h and then exposed to MeHg (2 μM) for 4 h. Insulin secretion stimulated by 20 mM glucose was detected by an insulin immunoassay kit. Data in A, B, D, and F are presented as mean ± SD of six independent experiments assayed in triplicate. * *p* < 0.05 compared to vehicle control; **^#^**
*p* < 0.05 compared to MeHg treatment alone. Results shown in C and E are representative images of at least three independent experiments. β-actin was used as loading control.

**Figure 6 ijms-23-02858-f006:**
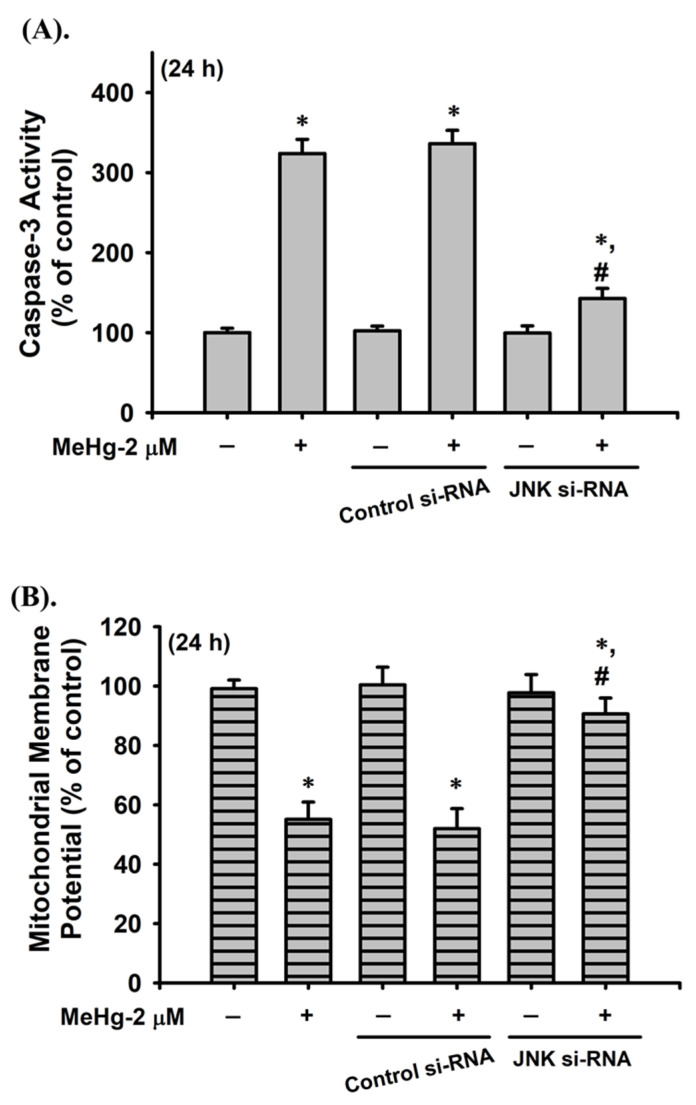
Transfection with siRNA-JNK prevented MeHg-induced β-cell apoptosis. RIN-m5F cells were transfected with control siRNA or siRNA specific for JNK prior to treatment with MeHg (2 μM). (**A**) Caspase-3 activity was measured using a Caspase-3 Activity Assay Kit. (**B**) Mitochondrial membrane potential (MMP) depolarization was detected using flow cytometry. (**C**) Protein expression of caspase-3, procaspase-12, CHOP, and XBP-1s and (**D**) phosphorylation of JNK1/2 protein were examined using Western blot analysis. Data in A and B are presented as mean ± SD of six independent experiments assayed in triplicate. * *p* < 0.05 compared to vehicle control; **^#^**
*p* < 0.05 compared to MeHg treatment alone. Results shown in C and D are representative images of at least three independent experiments. β-actin was used as loading control.

**Figure 7 ijms-23-02858-f007:**
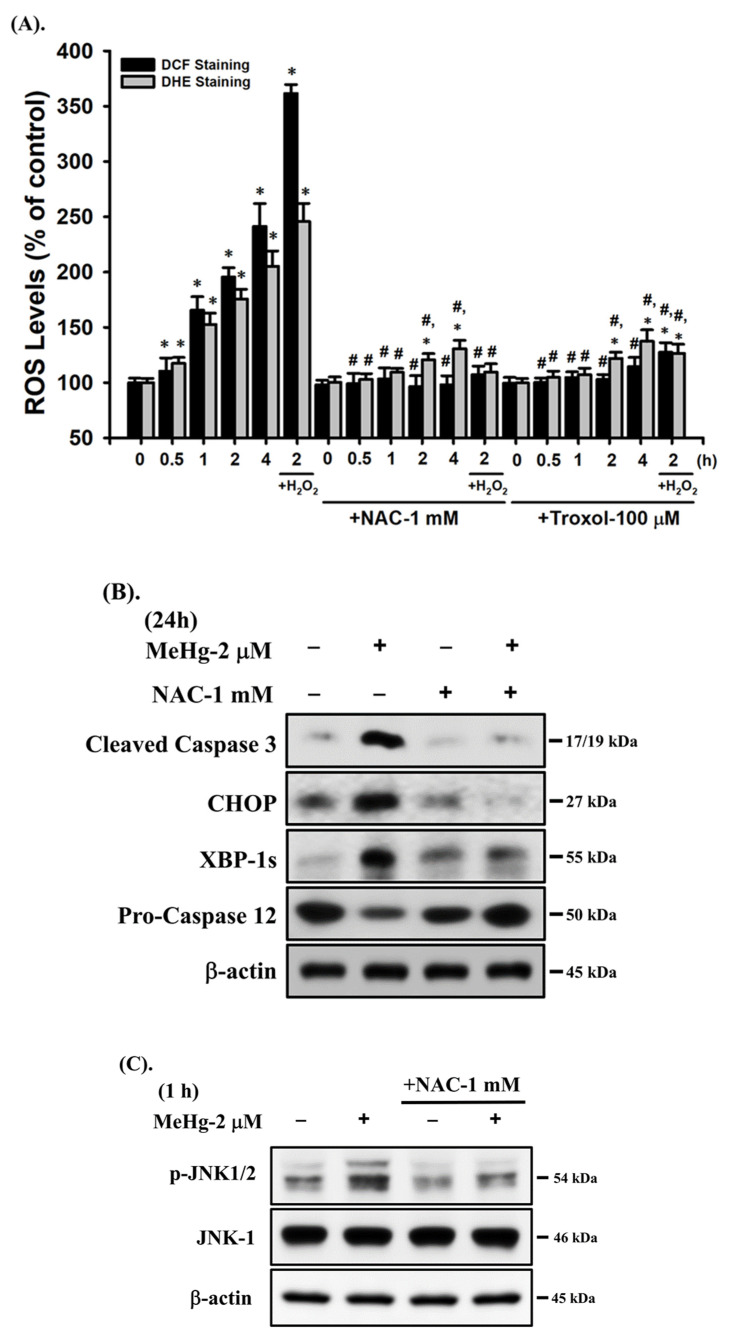
Role of ROS in MeHg-induced JNK-signaling activation and apoptosis in RIN-m5F cells. Cells were treated with MeHg (2 μM) for different time intervals in the absence or presence of NAC (1 mM) and Trolox (100 μM) for 1 h prior to addition of MeHg. (**A**) ROS generation was determined by flow cytometry using the fluorescent probes: DCF and DHE. H_2_O_2_ (50 μM) was used as a positive control. (**B**) Protein expression of caspase-3, procaspase-12, CHOP, and XBP-1s and (**C**) phosphorylation of JNK1/2 protein were examined using Western blot analysis. (**D**) In addition, RIN-m5F cells were treated with MeHg (2 μM) for 1h in the absence or presence of the specific inhibitor of JNK (SP600125; 10 μM) for 1 h prior to addition of MeHg. ROS generation was detected by flow cytometry using the fluorescent probe: DCF and DHE. Data in A and D are presented as mean ± SD of six independent experiments assayed in triplicate. * *p* < 0.05 compared to vehicle control; ^#^
*p* < 0.05 compared to MeHg treatment. Results shown in B and C are representative images of at least three independent experiments. β-actin was used as loading control.

## Data Availability

Please contact the corresponding author for reasonable data request.

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
