# Peer review of "Methylmercury Induces Mitochondria- and Endoplasmic Reticulum Stress-Dependent Pancreatic β-Cell Apoptosis via an Oxidative Stress-Mediated JNK Signaling Pathway"

_ijms, 2022, doi:10.3390/ijms23052858_

Round 1

Reviewer 1 Report

.The manuscript by Ching-Yao Yang et al entitled “Methylmercury induces mitochondria- and endoplasmic reticulum stress-dependent pancreatic β-cell apoptosis via an oxidative stress-mediated JNK signalling pathway” investigates the the effects of methylmercury in b-cell cytotoxicity.

The manuscript is consistent and properly organized. However certain issues should be addressed prior to publication.

- Could the panel organization in Fig2 be improved ?. It seems a bit confusing to see panel E next to panel C and later at the end the panel D.

- Line 215. When treating with MeHg, what is the implication for (GRP)78 and 94 protein expression levels  not being affected. Please clarify in the text manuscript.

- Panels (C) a. and (D). a. in Fig3 seem a bit unnecessary to me since they are repeated in b. for both (C) and (D) panels.

- could the authors specify what is the control for si-RNA in figure 3 ?

-Fig 5 (E).I guess treatment of MeHg +SP600125 is not indicated.

- The authors repeatedly indicate in the figure legends----representative images of at last three independent experiments--- why do the authors state “at last”?. Do they mean at least ???

- English in the discussion should be extensively revised and corrected.

Author Response

Reply to Reviewer’ 1 comments:

Q1. Could the panel organization in Fig2 be improved ?. It seems a bit confusing to see panel E next to panel C and later at the end the panel D.

Ans.: We have according to reviewer’s suggestion to reorganize the panels in Fig. 2.

Q2. Line 215. When treating with MeHg, what is the implication for (GRP)78 and 94 protein expression levels not being affected. Please clarify in the text manuscript.

Ans.: This sentence has been corrected and rewritten to ‘the levels of protein expression for glucose-regulated protein (GRP)78 and 94 were not changed by the MeHg treatment’ (in P6, L208-L210).

Q3. Panels (C) a. and (D). a. in Fig3 seem a bit unnecessary to me since they are repeated in b. for both (C) and (D) panels.

Ans.: We have according to reviewer’s suggestion to delete the panels (C)-a, and (D)-a in Fig. 3.

Q4. could the authors specify what is the control for si-RNA in figure 3 ?

Ans.:

  1. In Fig. 3C and D, ‘Control si-RNA’ was be transfected with ‘SignalSilence®Control siRNA (Cell Signaling Technology, Danvers, MA, USA)’, that was a siRNA sequence and would not lead to the specific degradation of any cellular message. Our results showed that transfection of RIN-m5F cells with control siRNA for 24 h did not change the protein expression of CHOP (Fig. 3C-a) and XBP-1u (Fig. 3D-a) as same as the vehicle control.
  2.  Furthermore, RIN-m5F cells were transfected with control siRNA prior to treatment with MeHg for 24, and then treated with 2 mM MeHg for 24 h. The results were showed the significantly induced protein expression levels of CHOP (Fig. 3C) and XBP-1s (Fig. 3D) in the MeHg-treated RIN-m5F cells, which was consistent with the normal (untransfected) RIN-m5F cells-treated with MeHg (2 μM) for 24 h (Fig. 3A).
  3. Therefore, these were the control for si-RNA in Fig. 3.

Q5. Fig 5 (E).I guess treatment of MeHg +SP600125 is not indicated.

Ans.: We have added the indicator ‘+’ in the treatment of MeHg + SP600125.

Q6. The authors repeatedly indicate in the figure legends----representative images of at last three independent experiments--- why do the authors state “at last”?. Do they mean at least ???

Ans.: This mistake has been corrected in the section of ‘Figure legends’.

Q7. English in the discussion should be extensively revised and corrected.

Ans.: Thank you very much for reviewer’s suggestion. The section of ‘3.Discussion’ has been rewritten and corrected carefully. We have also revised and proofreading our revised manuscript with language usage, grammar and syntax by a native English speaker. If further editorial corrections are needed, we will be pleased to revise according to your suggestions.

Reviewer 2 Report

The authors in this manuscript have presented  that oxidative stress-activated JNK signal plays a crucial role in the downstream regulation of both mitochondria-dependent and triggered apoptosis in MeHg-induced pancreatic β-cell death.

Comments:
1.Cytochrome c release is regulated byprotein-protein interactions between the Bcl-2 family proteins. The expression of  Bcl-2, Bcl-xL, Bax, Bid expression  is decrease or increase?

2. Provide explanation for choosing the glucose treatment concentrations at 2.8 mM  and 20mM .

3.  Provide explanation for choosing the NAC treatment concentrations at 1 mM.

Author Response

Reply to Reviewer’s 2 comments:      

Q1.Cytochrome c release is regulated by protein-protein interactions between the Bcl-2 family proteins. The expression of Bcl-2, Bcl-xL, Bax, Bid expression is decrease or increase?

Ans.:

  1. Bcl-2 family plays a crucial role in the regulation of cytochrome c release by functioning as inhibitors (such as Bcl-2) or promoters (such as Bax and Bak) of cell apoptosis. In order to investigate the effect of MeHg on Bcl-2 family, the levels of mRNA expression for Bcl-2, Bax, and Bak were detected by real-time quantitative RT-PCR.
  2. As shown in Figure 2C, the significant decrease in Bcl-2 (anti-apoptotic) and increase in Bax and Bak (pro-apoptotic) gene expression levels were revealed in RIN-m5F cells treated with 2μM MeHg for 24 h (p < 0.05).
  3. These results suggested that the levels of protein expression for Bcl-2 could be decreased, and for Bax and Bak could be increase in MeHg-treated RIN-m5F cells.

Q2. Provide explanation for choosing the glucose treatment concentrations at 2.8 mM and 20 mM .

Ans.:

  1. In health people, insulin release from b-cells is stimulated primarily by glucose present in the blood. As circulating glucose levels rise such as after ingesting a meal (blood glucose levels about 140-180 mg/dL (7.8-10 mM)). The blood glucose levels following fasting overnight is about 70-99 mg/dL (3.8-5.5 mM)).
  2. Thus, 2.8 mM (basal) and 20 mM (stimulatory) glucose concentration are choosing to examine the effect of insulin secretion function in RIN-m5F cells after MeHg treated, which was similar with other studies (Huang et al., 2021; Yang et al., 2016).

References:

Huang, C.C., Yang, C.Y., Su, C.C., Fang, K.M., Yen, C.C., Lin, C.T., Liu, J.M., Lee, K.I., Chen, Y.W., Liu, S.H., Huang, C.F. 4-Methyl-2,4-bis(4-hydroxyphenyl)pent-1-ene, a Major Active Metabolite of Bisphenol A, Triggers Pancreatic beta-Cell Death via a JNK/AMPKalpha Activation-Regulated Endoplasmic Reticulum Stress-Mediated Apoptotic Pathway. Int J Mol Sci 2021, 22. https://doi.org/10.3390/ijms22094379.https://doi.org/10.1016/j.tox.2019.152252.

Yang, T.Y., Yen, C.C., Lee, K.I., Su, C.C., Yang, C.Y., Wu, C.C., Hsieh, S.S., Ueng, K.C., Huang, C.F. Molybdenum induces pancreatic beta-cell dysfunction and apoptosis via interdependent of JNK and AMPK activation-regulated mitochondria-dependent and ER stress-triggered pathways. Toxicol Appl Pharmacol 2016, 294, 54-64. https://doi.org/10.1016/j.taap.2016.01.013.

Q3. Provide explanation for choosing the NAC treatment concentrations at 1 mM.

Ans.:

  1. In our preliminary study, the results found that the treatment of RIN-m5F cells with MeHg (1-4 μM) for 24 h in the absence or presence of NAC (1 mM) for 1 h prior to the addition of MeHg could effectively and fully increased the viable cells.

  1. However, pretreatment of RIN-m5F cells with NAC (0.5 mM) for 1 h prior to MeHg (1-4 μM) exposure also effectively reversed the decreased cell survival, but that was not fully reversed.
  2. Therefore, we chose the NAC treatment concentrations at 1 mM.

Figure 1. Effects of MeHg on cytotoxicity in RIN-m5F cells. Cells were treated with MeHg (1-4 μM) for 24 h in the absence or presence of NAC (1 and 0.5 mM) for 1 h prior to the addition of MeHg. Cell viability was detected using MTT assay. Data are presented as the means ± S.D. of six independent experiments assayed in triplicate. *p < 0.05 as compared to vehicle control. #p < 0.05 as compared to MeHg treatment alone.

Reviewer 3 Report

The paper deals with a molecular mechanism of methylmercury (MeHg)-induced pancreatic β-cell cytotoxicity. The authors interpret the data as reactive oxygen species (ROS)-activated JNK signaling is a crucial mechanism underlying MeHg-induced mitochondria- and ER stress-dependent apoptosis, ultimately leading to β-cell death.
The paper is written with sufficient clarity and contains some new data on the effects of MeHg in β-cells., which may benefit the scientific community. However, there are some pitfalls in the experimental design which may lead to misinterpretation of the data. These include the use of N-acetylcysteine (NAC) as an “antioxidant” and DCFH to measure ROS.
NAC, besides being a thiol-reducing compound, is also well known for its heavy metal chelating properties. Heavy metal chelators are known for their ability to reverse MeHg toxicity, and the data in Fig. 1A and 7A are most easily interpreted as the ability of NAC to chelate MeHg, which has nothing to do with the proposed antioxidant properties of NAC. The experiments would benefit from using another cell-permeable compound with antioxidant properties that does not chelate MeHg (such as Trolox or mitoQ).
Similar drawbacks can be met when using DCFH as the sole method to detect ROS production. The redox chemistry of DCFH is complex, and there are several limitations and artifacts associated with the DCF assay for intracellular ROS measurement. For example, cytochrome c, a heme protein, is capable of oxidizing DCFH directly or indirectly via a peroxidase-type mechanism, forming DCF. Also, redox-active metals promote DCFH oxidation in the presence of oxygen. Therefore, the data of DCF fluorescence may not reflect intracellular ROS production and may be misleading. It is generally recommended that DCFH is not the only method used to detect intracellular ROS production to avoid misinterpretation of the data. 

Author Response

Reply to Reviewer’ 3 comments:

Q1. These include the use of N-acetylcysteine (NAC) as an “antioxidant” and DCFH to measure ROS. NAC, besides being a thiol-reducing compound, is also well known for its heavy metal chelating properties. Heavy metal chelators are known for their ability to reverse MeHg toxicity, and the data in Fig. 1A and 7A are most easily interpreted as the ability of NAC to chelate MeHg, which has nothing to do with the proposed antioxidant properties of NAC. The experiments would benefit from using another cell-permeable compound with antioxidant properties that does not chelate MeHg (such as Trolox or mitoQ). Similar drawbacks can be met when using DCFH as the sole method to detect ROS production. The redox chemistry of DCFH is complex, and there are several limitations and artifacts associated with the DCF assay for intracellular ROS measurement. For example, cytochrome c, a heme protein, is capable of oxidizing DCFH directly or indirectly via a peroxidase-type mechanism, forming DCF. Also, redox-active metals promote DCFH oxidation in the presence of oxygen. Therefore, the data of DCF fluorescence may not reflect intracellular ROS production and may be misleading. It is generally recommended that DCFH is not the only method used to detect intracellular ROS production to avoid misinterpretation of the data.

Ans.: Thank you very much for reviewer’s comments and suggestions.

We have according to reviewer’s suggestion to add the experiments (including the using antioxidant-NAC and Trolox, and the fluorescent probes: 2′, 7′-dichlorofluorescin diacetate (DCFH-DA (to measure the intracellular hydroxyl, hydrogen peroxide and other ROS activity)) and dihydroethidium (DHE (a highly selective for O2•−)) to confirm the effect of MeHg-induced ROS generation-mediated toxicity in the section ‘‘2.5. Intracellular ROS production was an important event in MeHg-induced pancreatic β-cell injury’’ and in Fig. 1A and Fig. 7A. The results were showed as following:

‘We next examined the role of ROS in MeHg-induced β-cell injury. As show in Figure 7A, flow cytometric analysis showed that exposure of cells to 2 μM MeHg for 0.5-4 h triggered a significant increase in the intracellular ROS generation in a time-dependent manner. This MeHg-induced ROS generation as well as the decreased cell survival could be effectively reversed not only by 6-hydroxy-2,5,7,8-tetramethylchroman-2-carboxylic acid (Trolox, 100 μM) but also by antioxidant N-acetylcysteine (NAC, 1 mM) (Fig. 7A and Fig. 1A), indicating that ROS played an important regulator in MeHg-induced β-cell injury.’

Round 2

Reviewer 3 Report

The authors have added new sets of experiments and answered most of my comments. Nevertheless, the revised version still requires some editing.

Minor points:

The new results should be reflected in the abstract, and the new pretreatment using Trolox should be mentioned in the Methods section as well.

There is a mention in the Methods of treatment with Vitamin C, which seems to have no relevance to the described results.

Dihydroethidium is abbreviated as HDE in the legend to Figure 7. It is better abbreviated as DHE, and this should be corrected.

Besides these points, I will not raise more objections against this manuscript.

Author Response

Q1. The new results should be reflected in the abstract, and the new pretreatment using Trolox should be mentioned in the Methods section as well.

Ans.: The results and the new pretreatment using about ‘Trolox’ have added in the secretion of ‘Abstract’ and ‘4. Materials and Methods’.

Q2. There is a mention in the Methods of treatment with Vitamin C, which seems to have no relevance to the described results.

Ans.: This mistake has been corrected.

Q3. Dihydroethidium is abbreviated as HDE in the legend to Figure 7. It is better abbreviated as DHE, and this should be corrected.

Ans.: This mistake has been corrected to ‘DHE’ in the legend to Figure 7.
